# Enhancing the Properties of Liquid Crystal Polymers and Elastomers with Nano Magnetic Particles

**DOI:** 10.3390/ma17215273

**Published:** 2024-10-30

**Authors:** Sarah J. Reeves, Dil Patel, Peter J. F. Harris, Geoffrey R. Mitchell, Fred J. Davis

**Affiliations:** 1Polymer Science Centre, University of Reading, Whiteknights, Reading RG6 6AF, UKdil.patel@aru.ac.uk (D.P.); 2Centre for Advanced Microscopy, University of Reading, Whiteknights, Reading RG6 6AF, UK; p.j.f.harris@bristol.ac.uk; 3Centre for Rapid and Sustainable Product Development, Polytechnic of Leiria, 2430-080 Marinha Grande, Portugal; 4Department of Chemistry, University of Reading, Whiteknights, Reading RG6 6AD, UK; f.davis@reading.ac.uk

**Keywords:** liquid crystal, polymers, elastomers, ferromagnetic particles

## Abstract

Side-chain liquid crystal polymers have been mixed with ferromagnetic particles, and the formation of a monodomain in magnetic fields studied. At relatively low concentrations, the presence of ferroparticles substantially speeds up the rate of formation of a monodomain within the magnetic field, and, at a given concentration of ferroparticles, that rate is independent of the magnetic field’s strength. In this way, the rapid formation of a monodomain is possible at magnetic field strengths far lower those required for the liquid crystal polymer alone. This is anticipated to be very helpful in the fabrication of devices based on monodomain liquid crystal elastomers. Wide-angle x-ray scattering has been used to monitor the formation of the monodomain and small-angle x-ray scattering gives some indication of the ferroparticles’ behaviour. A model is developed to explain their behaviour. The alignment properties of the ferroparticles are related to their ability to form chains under the influence of very low magnetic fields; these chains are of relatively low stability and may become disrupted after long periods of time, high magnetic fields, or high concentrations. In general, the best results for alignment were at volume fractions below 1%, and under these conditions there is the potential for producing monodomain samples with improved properties; in particular, shape changes with temperature are significantly larger as a result of improved backbone orientation. Experiments involving monodomain formation and director realignment suggest that the presence of ferroparticles results in a modification of the mechanism for alignment development, driven by the organization of the polymer backbone, as a consequence of the constraints offered by the morphology of the chains of the ferroparticles.

## 1. Introduction

Liquid crystal elastomers are complex materials which combine the long-range orientational ordering properties of liquid crystals with network formation, providing materials with the potential for, for example, reversible deformation [1]. It has been shown by various authors that cross-linking liquid crystal polymers produces materials with a memory of their state at the time of cross-linking. This may result in the stabilization of, for example, the nematic phase through a reduction in phase transition temperatures or, for aligned samples, a memory effect [2]. In the memory effect, samples cross-linked following their alignment in, for example, a nematic phase can be heated to their isotropic phase, whereupon all the alignment of the liquid crystal phase is lost; on cooling below the nematic isotropic transition temperature, the original orientation is regained. This memory effect is entropically driven through the network’s elasticity, which naturally returns the polymer chain conformation to the arrangement present at the time of cross-linking. Perhaps even more striking is the chiral memory retained by the networks formed in the chiral nematic phase, where memory is still present when the chiral imbalance has been removed [3].

A consequence of the memory effect in liquid crystalline elastomers is that these materials can exhibit shape changes at a phase transition temperature. This shape change is a consequence of the alignment of the polymer backbone relative to the mesogenic groups [4]. The basic couplings which can be exhibited in a nematic system are shown in Figure 1. Guo et al. (1991) [5] demonstrated that these different couplings were possible using a single random copolymer system, in which the proportions of two monomers with different coupling chains were systematically varied and a composition was identified for zero coupling, which they termed N_0_, where the competing effects cancelled out. They used a simple mean field model to show that due to the restriction of the conformation of the coupling chain and the influence of the nematic field, the two components of the coupling had a differing impact on the behaviour of the system at a ratio of 2/1.5 to 1. Quantitative evidence of this coupling has been obtained using the neutron scattering of deuterium-labelled polymer chains [6,7]. Distortions of the polymer backbone at a microscopic level are then transmitted to the macroscopic level through the polymer network and thus, for a side-chain liquid crystal polymer in a liquid crystal state, polymer chains are distorted through coupling to mesogenic groups. Such behaviour has the potential to be developed into devices, such as actuators [8]; however, their maximum functionality requires the development of high levels of polymer chain orientation and this may be highly dependent on the way in which side-chain alignment is produced. 

There are a number of ways of inducing an alignment into a liquid crystalline polymer to form monodomain sample [2]. Generally, these can be categorized as alignment via an external electromagnetic field, alignment via a mechanical stress, and alignment via surface effects. An early example of the use of an external magnetic field is the method designed initially by Legge et al. [2], which involves the use of a magnetic field to induce a common alignment of the domain directors with subsequent cross-linking over a longer timescale. Probably the most well-known example of mechanical alignment is that devised by Küpher et al. [9], which relies on applying a uniaxial strain to a lightly cross-linked polymer (necessary to prevent the free flow of the material) followed by further cross-linking under load. Of these two examples, magnetic alignment is particularly convenient (as it does not require a pre-cross-linking stage or connection to the external force), but the latter method has shown the potential to form extremely well-ordered samples, with consequent potential for larger functionality, for example, in shape changes.

The two examples above impart different properties by virtue of their mode of alignment. For the mechanically aligned sample, the uniaxial strain imparts orientation to the polymer backbone, extending the chain; mesogenic alignment arises by virtue of coupling between the side chains and the polymer backbone. Dependent on the nature of the coupling chain, the mesogens will align either parallel or perpendicular to the draw direction. In contrast, with a magnetic field, the side groups align in response to the applied field; the polymer conformation will then reorganize to maintain the preferred relative orientation of the backbone and the side chain. It is crucial to the properties of the final material that significant cross-linking does not occur before alignment is complete.

In this work, the alignment of liquid crystalline polymers in the presence of ferro-nanoparticles is discussed [10]. We have recently shown that the introduction of surfactant-coated iron oxide nanoparticles can result in materials which show temperature-induced anisotropic shape changes which are consistently larger than those observed in the absence of these particles [11]. Here, we describe the alignment processes of the various components of this composite material and discuss how its design can be optimized to develop new actuator materials.

## 2. Materials and Methods

The liquid crystalline polymers prepared were copolymers of the aromatic ester [I] with hydroxyethyl acrylate [II], synthesized following the basic procedures of Portugall et al. [12] with modifications developed by Patel. Essentially, Portugall et al. synthesized the material by forming an acid chloride from thionyl chloride and 4-(8-propenoxyloxyhexoxy) benzoic acid. This methodology produces a side-product that is difficult to remove. The *Patel modification* takes this directly from an acid to the ester product, eliminating the requirement for the acid chloride and so avoiding a side-product. The method employed here used dicyclohexylcarbodiimide, which produced dicyclohexylurea as a side-product, which can easily be removed by filtration [13].

Polymerisation was initiated using 1 mol% azo-bis-iso-butyronitrile in a 10% *w*/*v* solution of the monomer in deoxygenated chlorobenzene. The reaction was performed under vacuum at 55 °C and left for 24 h. The products were characterized using infra-red spectroscopy ^1^H and ^13^C NMR (Nuclear Magnetic Resonance) and gel permeation chromatography, as described previously [13]. Phase behaviour was observed by DSC and optical microscopy. Nematic isotropic transition temperatures (T_NI_) were extremely sensitive to sample history; the values quoted were obtained from optical microscopy following slow heating to allow the equilibration of the polymer chain conformations [13] and thus obtain the equilibrium value of the transition temperature. DSC measurements become less sensitive as the scan rates are significantly lowered. The sample used for most of these studies is denoted CBZ6. See Table 1 for its characteristics.

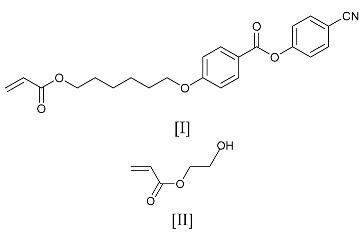



The cross-linking of the liquid crystal polymers to form elastomers was achieved by reacting the OH from the hydroxyethyl acrylate groups with a diisocyanate; in this case, 1-isocyanato-4-[(4-isocyanatocyclohexyl)methyl]cyclohexane (HMDI) [III]. This cross-linking agent was chosen as the rate of its reaction with the pendant OH groups is significantly slower than the rate of the orientation of the mesogens in the magnetic field. In a typical experiment, a quantity of CBZ6 (*ca.* 15 mg) was dissolved in dry dichloromethane (*ca.* 0.15 mL) and mixed with a solution of the cross-linking agent (*ca.* 4 mol%) in dry dichloromethane. A film was cast and the solvent allowed to evaporate. The polymer film was then heated briefly above its nematic isotropic transition temperature cooled to its cross-linking temperature and allowed to cure for several days.

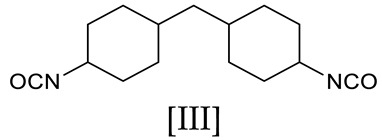


The dry ferrofluid was obtained from Ferrofluidics Ltd. (Bicester, Oxfordshire, UK). It consists of spherical surfactant-coated magnetite particles with a 10 nm diameter. This was confirmed by examination of the dry ferrofluid using a transmission electron microscope (Philips CM20 (Philips, Cambridge, UK)) with an accelerating voltage of 200 kV. Figure 2 shows an electron micrograph confirming the diameter of the ferroparticles. A ferroparticle-containing polymer film was prepared by mixing ferroparticles dispersed in dichloromethane with the polymer at various concentrations. Solutions were cast on an appropriate surface, typically Kapton, or, for microscopy, a glass slide.

The nematic phase exhibits long-range orientational order with a domain director and an order parameter S, where S is defined as in Equation (1) [14].
*S* = 1/2 <3*cos*^2^*α* − 1> (1)
where *α* is the angle between the long axis of the molecular unit—in this case the mesogenic side-group—and the director. In a typical sample prepared on an untreated surface, local surface effects will introduce variation in the orientation of the director, as schematically shown in Figure 3. It is the spatial variation of the director which gives rise to the characteristic optical textures which can be observed under an optical microscope.

Holding this polydomain film in a magnetic field will lead to a common alignment of the domains and the elimination of defects, leading to an optically transparent film and a monodomain sample. In the case of elastomer preparation, if the monodomain’s formation takes place sufficiently rapidly, then no significant cross-linking will have taken place. In this material, the 4-cyanophenyl 4-benzoate mesogen has a positive diamagnetic anisotropy and it will align with the longest axis parallel to the magnetic field. The time taken for monodomain formation depends on both the diamagnetic anisotropy Δ*χ* and the rotational viscosity *λ*_1_. Typically, the orientation time τ for an applied magnetic field of *H* is given by Equation (2) [15], where *μ*_0_ is the permeability of a vacuum.
(2)$$\tau =\frac{\lambda_{1}}{\mu_{0}\Delta \chi H^{2}}$$

As the mesogens rotate, the polymer backbone also undergoes reorientation. The simple model contained in Equation (2) may not be sufficient to completely account for the influence of the polymer chains on the alignment time, which include the effect of chain length on viscosity, but it provides us with a model to develop.

The formation of monodomains was performed using a Newport Instrument 4-inch electromagnet with 1.5-inch conical pole pieces. The magnet was water-cooled and the gap between the pole pieces was 10 mm, producing a maximum field of 2.1 T. The magnetic field was calibrated using a Hall probe and was homogenous over the sample area. The sample was mounted in a hot stage developed in-house connected to a 3-term PID temperature controller. The layout is shown in Figure 4a. In some situations, the state of the liquid crystal polymer/elastomer was probed using optical techniques, as shown schematically in Figure 4b.

Figure 5 shows a plot of the light measured through cross-polarizers upon aligning a liquid crystal polymer in a magnetic field. The intensity of the light reflects the global alignment of the mesogens, with maximum intensity reflecting maximum alignment; from these data a measure of the alignment rate can be obtained. As can be seen from Figure 5, there is clearly a significant temperature effect. This rate of alignment will depend on the sample’s temperature and the phase transition temperature (vide infra). To more readily quantify the level of alignment, wide-angle X-ray scattering measurements taken with a 3-circle diffractometer were used. 

For the measurement of shape changes after the formation of the monodomain, a small sample, approximately 1 mm by 1.5 mm, was cut from a monodomain CBZ6 elastomer with its longest length along the director. The sample was cut above the Tg, at 40 °C, to prevent the cracking that occurs in the brittle glass phase. This rectangular piece of elastomer was suspended in non-interacting silicon oil in a cavity slide and a cover slip was placed over the cavity. The shape changes were monitored using a video camera attached to the microscope hot stage system.

Optical microscopy was undertaken using a Carl Zeiss Jenalab polarizing microscope and samples were heated using a Linkam TH600 microscope hot stage (Linkam Co., Surrey, UK).

Wide-angle X-ray scattering (WAXS) measurements were performed using a transmission geometry three-circle diffraction system using the Cu Kα line at 1.54178 Å [16]. The geometry is shown in Figure 6. The X-ray beam was collimated to produce a 1 mm^2^ beam and the X-rays were detected by a scintillator detector. Using this system, the scattering can be measured over a |*Q*| range of 0.2 Å^−1^ to 6.5 Å^−1^ and from α = 0° to 360°, where |*Q*| = 4πsin(θ)/λ, 2θ is the scattering angle, and λ is the wavelength of the incident X-ray beam.

Small-angle X ray scattering (SAXS) experiments were performed using the Synchrotron Facility at the Daresbury Laboratory, Warrington, and, in particular, Beam Line 16.1. An X-ray wavelength of 1.41 Å and a RAPID 2-D detector were used. The sample-to-detector distance was 3.12 m, providing a |*Q*| range of 0.01 Å^−1^ to 0.14 Å^−1^. The detector was calibrated using the collagen from a wet rat’s tail; the SAXS pattern is a series of sharp orders occurring at regular intervals, for which the |*Q*| values are accurately known; this allows the pixels on the detector image to be converted to Q values [17].

## 3. Results

### 3.1. Phase Behaviour of Ferroparticles in a Liquid Crystal Polymer

Mixtures of the nematic polymer, CBZ6, with different quantities of ferroparticles were prepared in the way described above and observed under a polarizing optical microscope. Selected images are shown in Figure 7; as the ferroparticle content was increased to around 1.42% by volume, black objects were observed in the mixture (Figure 7a). As the ferroparticle content was increased further, the quantity of these objects was increased and the birefringence decreased; at around 5.68% by volume of added ferroparticles, a black network was observed in the liquid crystal texture (Figure 7c). These dark objects were also observed in the isotropic phase and were unchanged in the presence of a magnetic field, although here the birefringence became directional if the polymer was in its nematic phase.

At low concentrations of ferroparticles, the liquid crystal texture appears unchanged; for example, at 1% there appears to be no phase separation. At higher concentrations phase separation occurs; the ferroparticles formed aggregates in the liquid crystal polymer matrix. At the highest concentrations examined, the ferroparticles formed a network structure within the liquid crystal polymer, which became denser with increasing concentration. This network’s formation was related to a decrease in the birefringence.

The influence of the ferroparticles on the phase behaviour was further monitored by measurement of the nematic isotropic transition temperature; the data are presented in Figure 8. All the samples were prepared by casting films from dichloromethane and so are directly comparable. Small quantities of ferroparticles added to CBZ6 increased its T_NI_ by up to 2 °C. The addition of ferroparticles has also been reported to increase the smectic A-to-nematic transition temperature in another liquid crystal system [18]. The behaviour of this system is broadly in line with theoretical expectations [19]. Once a plateau had been reached, the T_NI_ was reduced by the addition of further ferroparticles; thus, a small quantity of ferroparticles stabilizes the liquid crystal phase, but once a certain concentration is reached it acts as an impurity and starts to reduce the stability of the liquid crystal phase, causing some phase separation.

### 3.2. Monodomain Formation

The alignment of the sample to form a monodomain can be observed both from light intensity measurements (see Figure 5 above) and from WAXS measurements. Figure 9 shows a plot of the intensity of X-ray scattering recorded parallel and perpendicular to the monodomain director or the direction of the applied magnetic field. The intense broad peak at Q~1.44 Å^−1^ arises from the short-range correlations between the mesogenic side chains, and the azimuthal variation in this peak is used to evaluate the global orientation parameter <P_2_>, which is the product of the nematic order parameter S and the domain director orientation parameter <D_2_> [20]. This process of quantitative evaluation is described in [21,22] and is a mathematically robust procedure which is summarized below. The scattering pattern I(|*Q*|, *α*) is expressed as a series of amplitudes of spherical harmonics I*_2n_*(|*Q*|), where only the even harmonics are required due to the presence of an inversion centre in the scattering pattern [21]. The amplitude of each harmonic is evaluated using Equation (3) [21].
(3)$$I_{2n}\left(\left|Q\right|\right)=\left(4n+1\right)\cdot {\int_{0}^{\frac{\pi }{2}}I\left(\left|Q\right|,\alpha \right)P}_{2n}\left(cos\alpha \right)sin\alpha \;d\alpha$$
where *P_2n_*(*cos α*) are a series of Legendre polynomials. The global orientation parameters <*P_2n_*(*cos α*)> can be found at a |*Q*| value corresponding to the maximum intensity of the structural element of interest using Equation (4)
(4)$$<P_{2n}\left(cos\alpha \right)>=\frac{I_{2n}\left(\left|Q\right|\right)}{I_{0}\left(\left|Q\right|\right)\left(4n+1\right)P_{2n\;}^{m}(cos\alpha )}$$
where $P_{2n\;}^{m}$ are the values calculated for the scattering from a perfectly aligned structure. Reference [22] gives some values for common structures. In this study, the scattering arising from the correlations between the mesogenic side groups was used, which gives a peak in the scattering at |*Q*|~1.44 Å^−1^ which intensifies in the equatorial section. The advantage of using this feature is that it is a more intense feature in the scattering pattern and it is well separated from other structural features.

Figure 10 shows a plot of the I(α) at a fixed value of |*Q*|= 1.44 Å^−1^, which is used to evaluate the global orientation parameters <*P*_2*n*_>. The plot shows the symmetrical curve which is inherent to the scattering data for a weakly absorbing sample such as CBZ6. The intensification of the peak at 90° and 270° directly indicates that the long axis of the mesogenic groups lies parallel to the magnetic field direction, as is expected for mesogens which exhibit a positive diamagnetic anisotropy. A more complete structural analysis is given in Reference [16].

Figure 11 shows the plot of the global orientation parameter against time in a magnetic field of 2.1 T. As can be seen, there is a rapid increase in orientation followed by a more gradual increase to eventually reach a steady value; by which time the cross-linking reaction is complete. From this it is possible to identify a value of time at which the orientation value reaches 50% of its final value (the alignment time) and this provides us with a route to understanding the factors which influence the rate of alignment. 

Figure 12 shows the variation in the alignment times as a function of the applied magnetic field strength. The plot shows the strong effect that the magnetic field strength has on the alignment time. The magnetic fields used in this work are at their limit due to the saturation of the magnetic field in the iron core pole pieces. Larger magnetic fields are available with air-cored magnetics, as in the case of superconducting magnets, where fields up to 45 T are possible. Clearly, heating the sample in a cryogenically cooled bore is more challenging than the methodology presented here. It is interesting that the alignment times follow the 1/B^2^ model (Equation (2)).

On the basis of our previous experience forming monodomains [11], it was expected that that the molecular weight would have a strong influence on the alignment times. Figure 13 shows the global orientation parameter <P_2_> plotted against time in a fixed magnetic field of 2.1 T and at 90 °C for two liquid crystalline polymers, CBZ6 (SIP 198) and SIP 127, with differing molecular weights but that are otherwise chemically equivalent (see Table 1). The lower-molecular-weight sample aligns much more rapidly. These differences can be attributed to the greater viscosity of the higher-molecular-weight polymer. The ratio of their alignment times is more than a factor of 16.

The techniques used to probe the orientation of the liquid crystal polymers were also used to evaluate the behaviour of mixtures of the polymer with ferro nanoparticles. Wide-angle X-ray scattering was used to probe their intermediate-range structure (distances between neighbouring molecules). The curves in Figure 14 are essentially the same as those shown in Figure 9 for the polymer, apart from the superposition of some small sharp Bragg reflections arising from the magnetite in the nanoparticles. It is clear from the comparison of the curves in the parallel and perpendicular sections that the magnetite scattering is isotropic. This was confirmed by azimuthual sections at peak positions for the magnetite [23] (for example, at |*Q*| = 2.1 Å^−1^(220), |*Q*| = 2.48 Å^−1^(311), |*Q*| = 3.89 Å^−1^(511), and |*Q*| = 4.26 Å^−1^(440)). The background in these sections is higher than the polymer alone due to the X-ray fluorescence of the iron. For these samples, the Kapton substrate used in their fabrication before measurement of their X-ray scattering has not been removed and so this also contributes to the background, and this is also shown in Figure 14.

Figure 15 shows the development of the global orientation parameter <P_2_> over time for samples of the polymer plus a small fraction of nanoparticles. A comparison of the two curves in Figure 15 shows a remarkable change in the alignment time with the addition of ferro nanoparticles. The time taken to reach 90% of the maximum value of <P_2_> observed was reduced by a factor of 4 by the addition of 0.28% *v*/*v* of ferroparticles.

In order to develop an understanding of the mechanism of action and the optimum fraction of ferroparticles, a range of polymer films with different compositions, ranging from 0.07% to 28.4% of the inorganic material, were prepared and their alignment time evaluated through repeated X-ray scattering measurements. Figure 16 shows the maximum value of <P_2_> for each composition held in a magnetic field of 2.1 T at a temperature of 110 °C. There are two regions to this plot. At lower concentrations of ferroparticles, the value of <P_2_> appears to exhibit a more or less constant value; the second region shows a rapidly reducing value above a concentration of 0.52%. In the first region, the ferroparticles appear to enhance or at least stabilize the maximum value of <P_2_>, whereas above a volume fraction of 0.52%, orientational order appears inhibited.

Figure 17 shows a plot of the alignment times of the monodomains formed from a mixture of the polymer CBZ6 and a small fraction of ferroparticles; remarkably, the alignment time is independent of the applied magnetic field. Note that for the mixture with 0.28% ferroparticles, this invariance could be observed at a very low field of ~0.15 T, this was the lowest field which could be obtained from the electromagnet used in the preparation system due to the remanent magnetisation of the iron pole pieces. To emphasize this remarkable observation, the alignment time τ is invariant with the magnetic field from 15 T to 2.1 T. This is in marked contrast to the alignment times for the monodomains prepared from the polymer, which show a strong dependence on the strength of the applied magnetic field, as shown in Figure 12.

The first proposal of mixing ferroparticles with liquid crystals was made by de Gennes and Brochard in a paper published in 1970 [24]. They suggested that there was both a magnetic field effect on liquid crystal alignment due to the anisotropic magnetic field surrounding even a spherical particle and a mechanical effect due to the elongated particle that they used. The invariant nature of the alignment time with the magnetic field in the current work suggests that there is indeed another alignment mechanism in place which is strong enough to mask the inherent magnetic field effect due to the diamagnetic anisotropy of the mesogenic units.

Ferrofluids, essentially a suspension of magnetic particles in a carrier fluid, were first developed in the 1960s and subsequently developed for a wide variety of applications [25,26], ranging from rocket fluid [26] through to magnetic hyperthermia in medical physics [27] and to fluid seals [28]. Magnetic polymer composites offer the potential for the controlled release of bioactive materials; for example, as coatings on implants [29]. Due to their size, nanoparticles are readily magnetized but usually have no magnetization when the applied magnetic field is removed; consequently, they readily form dipolar chains of nanoparticles [30]. Zubarev et al. [19] have developed a theoretical model (which predicts the mean number of particles $n_{\infty }$ in a chain due magneto-dipole inter-particle interactions as shown in Equation (5):(5)$$n_{\infty }=[1-2/3\;(\phi /\lambda^{2})e^{2\lambda }{]}^{-1}$$
where *ϕ* is the solid fraction and *λ* is the dimensionless coupling coefficient, which is a measure of the strength of particle–particle interactions and determines the probability of aggregate formation due to magneto-dipole interactions. *λ* can be derived using Equation (6):(6)$$\lambda =\frac{\mu_{0}M_{d}^{2}V}{24kT}\;$$
where *M_d_* is the saturation moment of the bulk, *V* is the particle volume, and *μ*_0_ is the permeability of the free space. To give an example, for a 10 nm magnetite particle at 298 K with *M_d_* = 466 kAm^−1^, n = 1.36, so little chaining occurs. However, for a 13 nm particle n is infinite. In this situation *λ*~0.4 and this is the regime under which long chains are formed. A *λ* ≥ 3 is required for spherical aggregates. At higher concentrations, Dubois et al. [31] observed a cellular-like structure.

It is to be expected that the chains of ferroparticles form parallel to the magnetic field and present a series of internal surfaces in a similar manner to polymer stabilized liquid crystals [32]. Just as the interactions with any surface will affect the alignment of the nematic domains, we propose a simple model to explain the results in Figure 17. In the absence of an external magnetic field, the ferroparticles are randomly arranged. Realistically, it is difficult to place the sample in a zero magnetic field due to the earth’s magnetic field of ~30 μT. Increasing the applied magnetic field to 0.15 T is sufficient to align the chains of the nanoparticles; the response time will depend on the threshold magnetic field, the viscosity of the polymer, and the number of internal surfaces. The latter gives rise to the composition dependence shown in Figure 17. Figure 16 shows the dependence of the maximum value of <P_2_> observed during the formation of the monodomain. Clearly, all of the compositions in the first region of this plot are sufficient to drive the domain alignment towards a monodomain, and we attribute this to the fact that these chains are organized in a parallel 2D manner. As the proportion of ferroparticles increases, the morphology of the chains becomes more 3D [30] and starts to inhibit monodomain formation.

In order to explore whether there is any additional evidence to support this chain model of the invariance of the monodomain formation time, both laser and X-ray scattering were used to probe the appropriate samples. A dispersion of ferroparticles in kerosene readily yields a highly anisotropic light scattering pattern between crossed polarizers lying normal to the applied magnetic field direction. The characteristics of the light scattering yielded objects 24 µm long and 5 µm wide. The external surface of a cross-linked sample of the polymer with a 0.28% volume fraction of ferroparticles showed a rippled surface when examined under a scanning electron microscope, with ripples which were µm wide. Although it is tempting to associate these with the chains of particles, the work of Zubarev et al. [19] predicts this rippled surface as the ferroparticles generate such a surface according to their field lines.

Figure 18a shows that the 2D small-angle scattering pattern of the monodomain shows strong scattering clustered around the zero-angle point (see also Appendix A). Although much of the scattering is obscured by the beam stop, it can be seen that it shows some highly anisotropic scattering, in which the scattering is highly constrained in the vertical direction, indicating some highly extended objects, and spread out in the horizontal direction, indicating that the extended objects are quite narrow in the horizontal direction. The analysis of these data using a log–log plot of the intensity vs. the scattering vector (Figure 18b) shows at a small q there is a linear relationship with a slope close to (-)1. This is suggestive of a one-dimensional structure [33], and we believe such features to arise from individual chains of ferroparticles. The strength of the scattering increases with an increasing volume fraction of the ferroparticles and an increasing level of anisotropy, and with increasing time in the magnetic field. The tentative analysis of these features by fitting a Gaussian peak to the observable data yields a width of the anisotropic object of 195 nm, with the length evaluation probably limited by the resolution of the setup but in excess of 1000 nm. Further information and SAXS data can be seen in the Appendix A.

### 3.3. Monodomain Shape Changes 

An aligned liquid crystal polymer cross-linked in a magnetic field produces a monodomain sample which will retain a memory of its alignment; in addition, the monodomain will change shape with temperature. In the case of the rectangular samples used here, and as their alignment is unidirectional and the sample has axial symmetry, when then viewed under a microscope (effectively in two dimensions), the sample was seen to become longer in one direction and shorter in the other. For the liquid crystal polymer containing 0.28% ferroparticles, it was found that at 40 °C, the length of the sample parallel to the alignment direction increased by 20% compared to the sample in its isotropic state (the perpendicular direction reduced in size by half this amount). This was substantially higher than the change in the elastomer with no ferroparticles; here, the maximum elongation was 16%. A full plot showing the sample shape as a function of temperature is provided in Figure 19.

Video recordings of the temperature-induced shape changes showed that the change in dimension was complete within 100 s following a temperature jump. The thermally induced shape change in the monodomain liquid crystalline elastomer reflects the orientation of the polymer backbone at the time of cross-linking. When the liquid crystal side groups align, the polymer backbone locally takes up a preferred orientation to the side chains (in this case parallel). This leads to a reorganization of the polymer backbone to reflect its preferred orientation. Subsequent cross-linking increases the stability of this orientation; the nematic phase is stabilized and a distortion of the polymer chain from this conformation at the time of cross-linking requires additional energy to overcome the elasticity of the polymer network. For a lightly cross-linked sample, heating to its isotropic phase results in a loss of any substantial orientational order (albeit with some persistent birefringence), and the polymer backbone takes up an isotropic arrangement. Since cross-linking in a monodomain avoids the masking effect of a polydomain morphology, microscopic shape changes are observable on a macroscopic level; thus, changes in backbone orientation are reflected in the overall shape of the polymer. Clearly, this method provides a more accessible technique for measuring the shape of the polymer chain trajectory than the multiple steps of preparing a selectively deuterated polymer and the subsequent small-angle neutron scattering described in Reference [7].

The observations described above show that the presence of nanoparticles of magnetite imparts a greater level of backbone orientation to the monodomain elastomer, albeit with a similar level of alignment of the mesogenic side groups. There are a number of possible origins for this effect, these include, firstly, an increase in local magnetic field strength; secondly, the ferromagnetic particles’ alignment induces more rapid mesogen alignment; thirdly, the formation of ferromagnetic chains induces the alignment of the polymer backbone. The alignment time was found was to be independent of field strength and occurred rapidly at low magnetic fields, so a mechanism involving an increase in the local field mechanism is unlikely. The fact that the ferroparticles are spherical, and the low concentration required for alignment, suggests that a torque mechanism of the type described by Brochard and de Gennes [24] is unlikely. From the observations described above, it can be seen that the ferroparticles form chains when a magnetic field is applied and it is these chains that are associated with a faster mesogenic unit alignment and, by implication, the polymer backbone. It seems likely that the mesogenic unit and the ferroparticle chains interact, particularly as chains have been observed when a magnetic field of 0.01 T was applied to a ferronematic polymer in its isotropic phase and then cooled into its liquid crystal phase. Crucial to this process is the rearrangement of the polymer backbone to optimize its preferential orientation relative to the mesogenic units; this process is likely to be substantially slowed by the need to break up local mesogenic order on a microscopic scale. The direct distortion of the polymer backbone is likely to be a more kinetically favoured process. The substantially increased backbone orientation evident from these samples does suggest that, at least in part, the chains formed by ferroparticles act directly on the polymer backbone.

### 3.4. Director Rotation 

In order to further understand the role of ferroparticles in the orientation of liquid crystal polymers, we conducted experiments to reorientate the aligned polymer systems. A 2.1 T magnetic field was applied at 90° to the director of an aligned ferronematic polymer (with a 0.28% volume fraction) and the change in its mesogenic side-group orientation was monitored using X-ray scattering with a three-circle diffractometer. The angle of the side-group orientation gradually changed to correspond to the new magnetic field direction over approximately 30 min. This time appears to be independent of the magnetic field strength at the fields investigated (0.8 T–2.1 T). It was also found that this reorientation resulted in a decrease in the global orientation parameter as the mesogens rotated; this decrease is independent of the field strength both in depth and time. The orientation level is recovered at the same time as the new angle of orientation is reached. This behaviour can be compared with the reorientation of CBZ6 without ferroparticles. Figure 15 shows that this system takes longer than the ferronematic polymer to reorganize (as with its initial monodomain formation). The intermediate decrease in orientation is much larger and there is a discontinuity in the change in orientation angle.

The difference in behaviour can be seen more clearly by looking at the X-ray scattering azimuthal scans, which show intensity as a function of alpha, where alpha is their angle to the original director. For the non-ferronematic polymer, Figure 20a shows a gap in the plots where the maximum orientation angle discontinuity occurs. In contrast, for the ferronematic polymer, Figure 20b shows a continuous array of plots where the orientation angle changes smoothly to the applied magnetic field direction.

As discussed above, the non-ferronematic polymer exhibits a discontinuity in its maximum orientation angle and a decrease in its level of orientation as director rotation occurs. This suggests that the mesogens are first unaligned and then realigned in the new direction, whereas the ferronematic polymer exhibits a continuous change in maximum orientational angle and only a small decrease in its level of orientation. Thus, for the ferronematic sample, it appears that the mesogens rotate together to the new direction, and this appears to a be a collective behaviour. This strikingly different behaviour lends itself to two potential explanations: The first one is that it is more energetically favourable for the ferroparticle chains to rotate to the new magnetic field direction rather than dissolve and re-assemble; since they influence the mesogens, the director follows the same behaviour. However, as we have seen above, the ferroparticle chains are particularly unstable and lose their alignment relatively easily, even without the additional influence of a change in the magnetic field direction. An alternative explanation is that for materials aligned in the presence of ferroparticles, it is a general realignment of the polymer backbone which is occurring; in this case, a loss of local mesogen alignment is energetically unfavourable, since it will require a greater distortion of the elastic network. This arises because the polymer backbone was originally better aligned with the ferroparticles and the backbone as a whole reorientates.

Figure 21 shows the shape change of an initially circular sample which was cut from a monodomain sample held in an isotropic phase. We have measured the dimensions of the sample at specific angular positions, a process easily achieved with digital imaging, as the sample temperature was varied. This approach also compensates for any error in aligning the sample with respect to the original magnetic field direction. As Figure 21 shows, the maximum change was observed at an angle slightly titled from the horizontal. This measurement was performed with the chemically identical polymer SIP 136 (see Table 1), which has a lower molecular weight.

## 4. Conclusions

Nanoscale spherical ferroparticles can be added to a side-chain liquid crystal polymer to form materials with novel properties. Low concentrations of these magnetically active materials significantly improve the alignment time for the formation of a monodomain sample. We expect that there will be a critical volume fraction of nanoparticles for this behaviour.

For ferroparticle concentrations above 1%, there is substantial phase separation of the system’s inorganic component. The action of a magnetic field induces the rapid formation of a chain of ferroparticles; such chains can be rather long on a molecular scale, with lengths of a number of microns. However, the chains are relatively unstable and become disrupted over time and by large magnetic fields, and at higher concentrations they are less likely to form or more likely to reorientate.

It has been seen that the formation of these structures is associated with enhanced monodomain samples. The properties of the monodomains from such materials are enhanced compared to liquid crystalline monodomains prepared in the absence of such ferrofluids; in particular, larger shape changes are observed as a function of temperature as the material is heated from its isotropic to its nematic phase. Experiments involving monodomain formation and director reorganization suggest that in the presence of ferromagnetic nanoparticles, the mechanism of monodomain formation is changed and driven by the polymer backbone.

## Figures and Tables

**Figure 1 materials-17-05273-f001:**
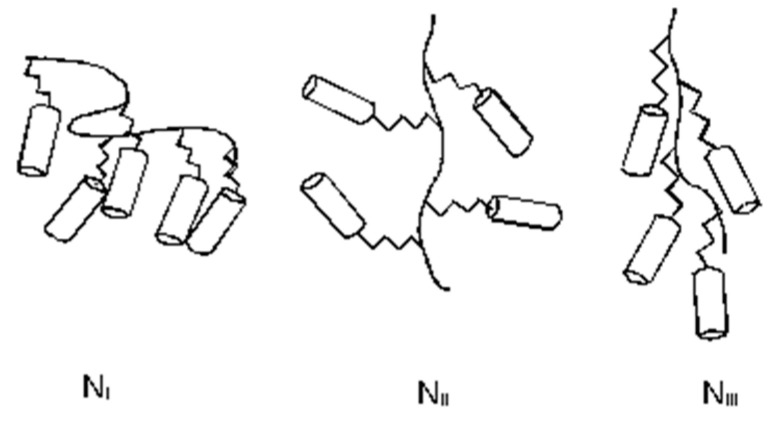
Molecular configurations of the side chain relative to the polymer backbone in a side-chain liquid crystal polymer. N_I_ and N_II_ arise due to a perpendicular orientation between the two species, forcing the backbone to become an oblate spheroid. A N_III_ phase arises when the orientation is parallel, forcing the polymer coil to become a prolate spheroid.

**Figure 2 materials-17-05273-f002:**
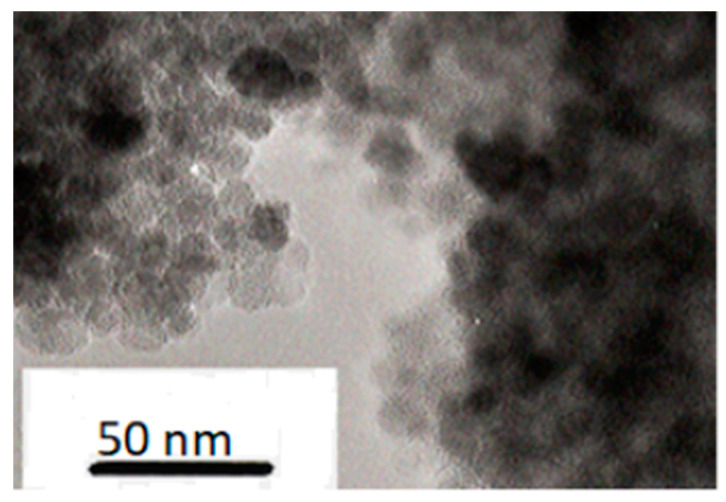
A transmission electron microscope image of the dry ferrofluid.

**Figure 3 materials-17-05273-f003:**
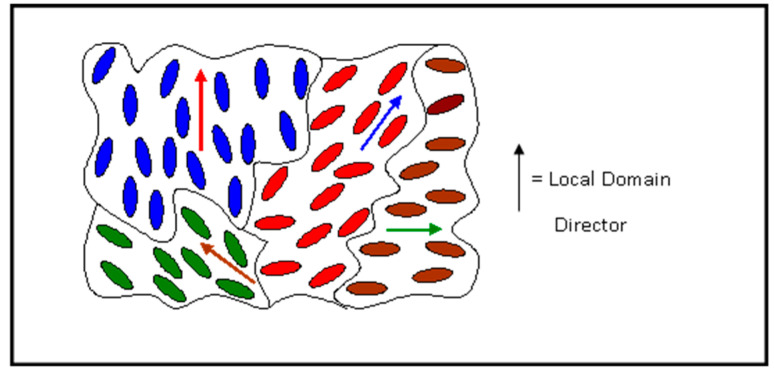
Domain structure of a nematic liquid crystal.

**Figure 4 materials-17-05273-f004:**
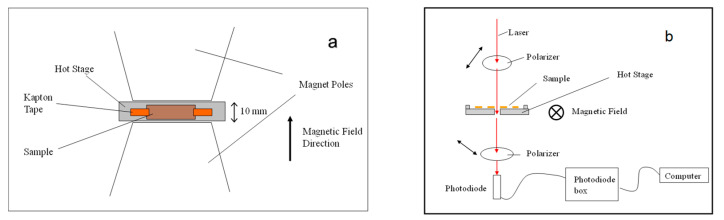
(**a**) Schematic diagram of monodomain formation apparatus viewed from above. (**b**) Schematic diagram of optical system used to monitor monodomain formation.

**Figure 5 materials-17-05273-f005:**
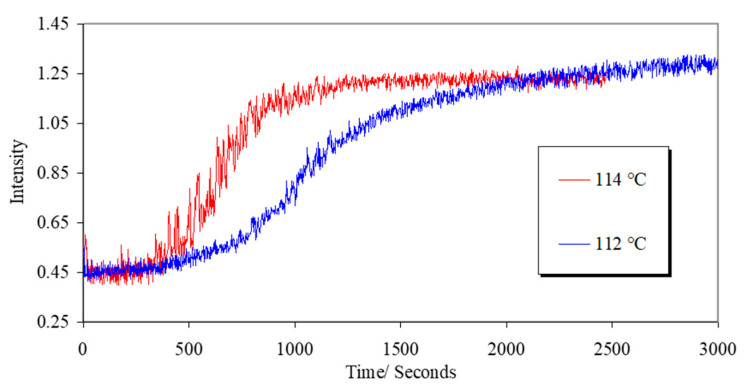
A plot of the measured light intensity of a liquid crystal polymer CBZ6 aligning with a magnetic field using the system shown in Figure 4b and at the temperatures indicated.

**Figure 6 materials-17-05273-f006:**
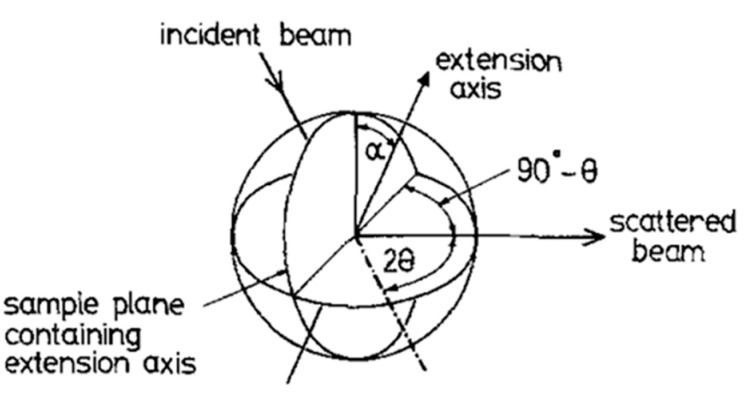
The geometry of the 3-circle transmission diffractometer. Adapted with permission from Ref. [16]. Copyright (1987), G.R Mitchell, F.J. Davis, A. Ashman.

**Figure 7 materials-17-05273-f007:**
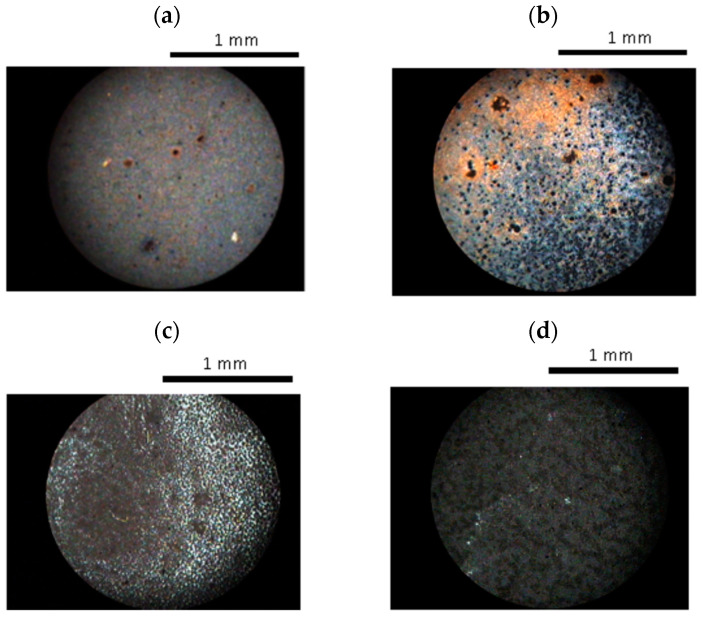
Optical microscope images (with crossed polarizers) of different volume fractions of ferroparticles in CBZ6: (**a**) 1.42%, (**b**) 2.84%, (**c**) 5.68%, and (**d**) 28.4%.

**Figure 8 materials-17-05273-f008:**
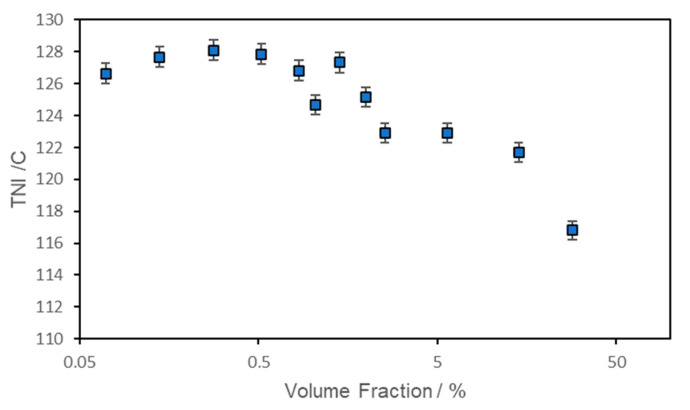
Nematic isotropic transition temperature obtained from optical microscopy as a function of the volume fraction of ferroparticles.

**Figure 9 materials-17-05273-f009:**
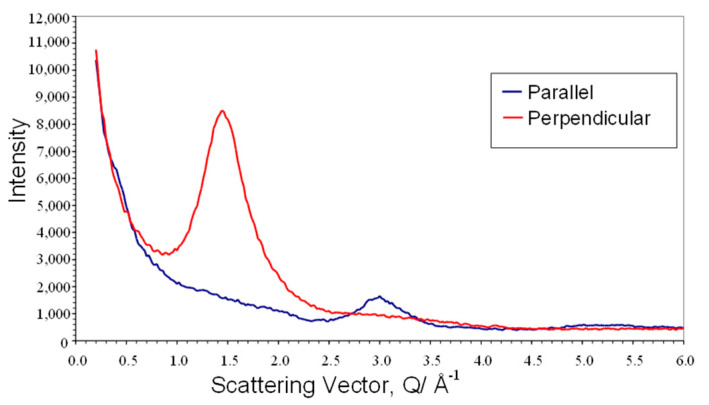
A plot of sections of the scattering data I(Q,α) plotted against the modulus of the scattering vector Q, for a monodomain sample of the liquid crystal polymer CBZ6 prepared at 90 °C and in a magnetic field of 2.1 T. The sample’s alignment is that its magnetic field direction is parallel to α = 0°.

**Figure 10 materials-17-05273-f010:**
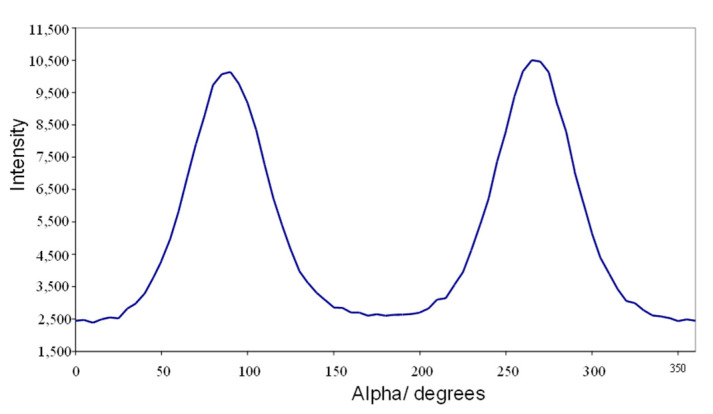
A plot of the global orientation <P_2_> against time of a sample of the liquid crystal polymer, CBZ6, used in this work, at 90 °C and in a magnetic field of 2.1 T.

**Figure 11 materials-17-05273-f011:**
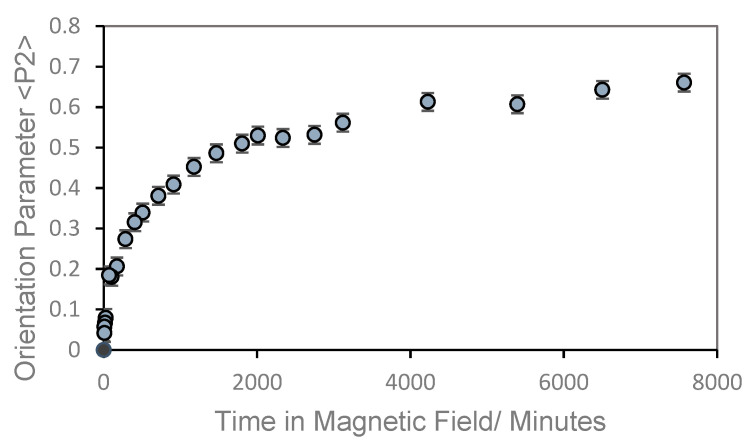
A plot of the global orientation <P_2_> against time of a sample of the liquid crystal polymer, CBZ6, used in this work, at 90 °C and in a magnetic field of 2.1 T.

**Figure 12 materials-17-05273-f012:**
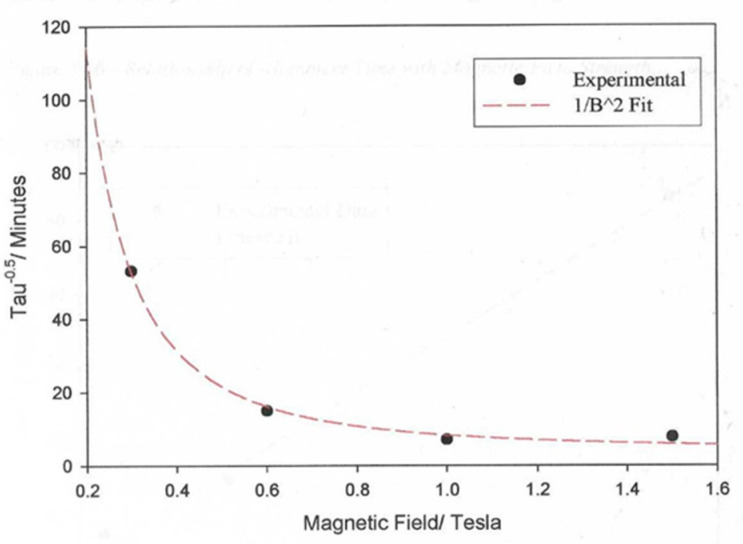
A plot of the values of τ 0.5 against the applied magnetic field obtained for a sample of the liquid crystal polymer, CBZ6, used in this work, at 90 °C and in a magnetic field of 2.1 T. The dashed line corresponds to the best fit to a model of the alignment times described in the text.

**Figure 13 materials-17-05273-f013:**
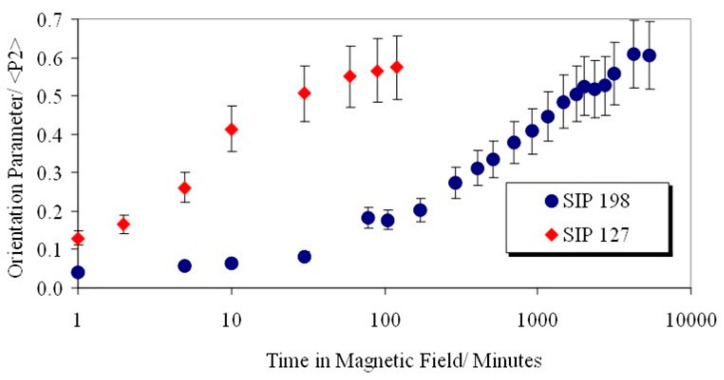
A plot of the global orientation <P_2_> against time of two sample of the liquid crystal polymer used in this work, at 90 °C and in a magnetic field of 2.1 T.

**Figure 14 materials-17-05273-f014:**
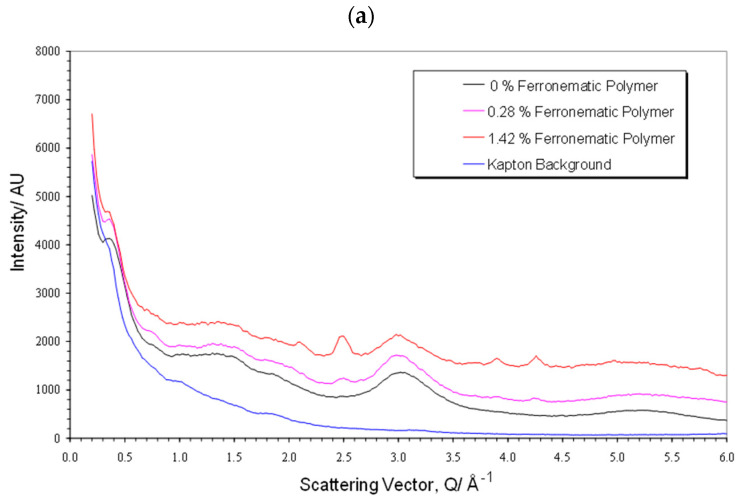
Wide-angle X-ray scattering scans of (**a**) samples of CBZ6, with varying concentrations of ferroparticles, as indicated in the index insert, parallel and (**b**) perpendicular to the aligning field, compared to the polymer alone.

**Figure 15 materials-17-05273-f015:**
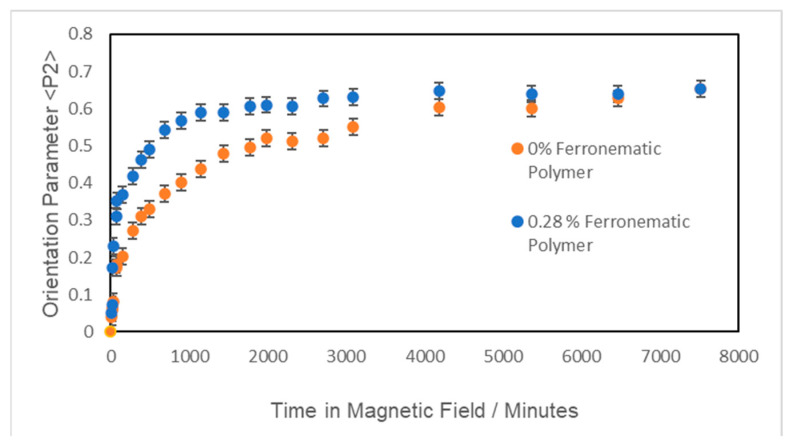
A plot of the global orientation <P_2_> against time of a sample of the liquid crystal polymer, CBZ6, used in this work and of a polymer with a 0.28% volume of nanoparticles, at 90 °C and in a magnetic field of 2.1 T.

**Figure 16 materials-17-05273-f016:**
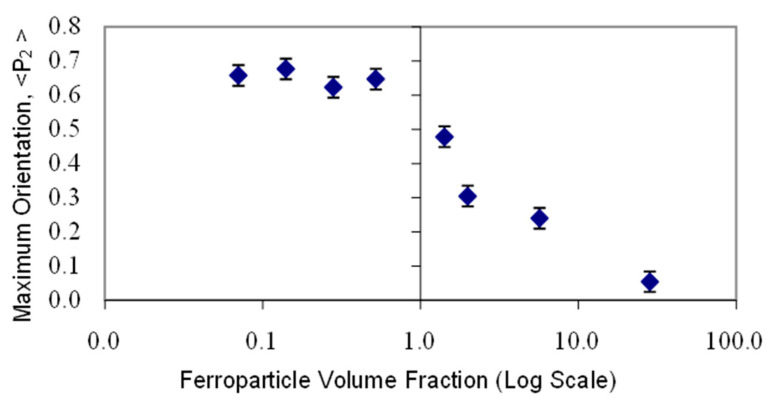
A plot of the values of the maximum global orientational parameter <P_2_> recorded for the side-chain liquid crystal polymer CBZ6 plus varying fractions of ferro nanoparticles in samples held at 110 °C and in a magnetic field of 2.1 T for substantial periods of time.

**Figure 17 materials-17-05273-f017:**
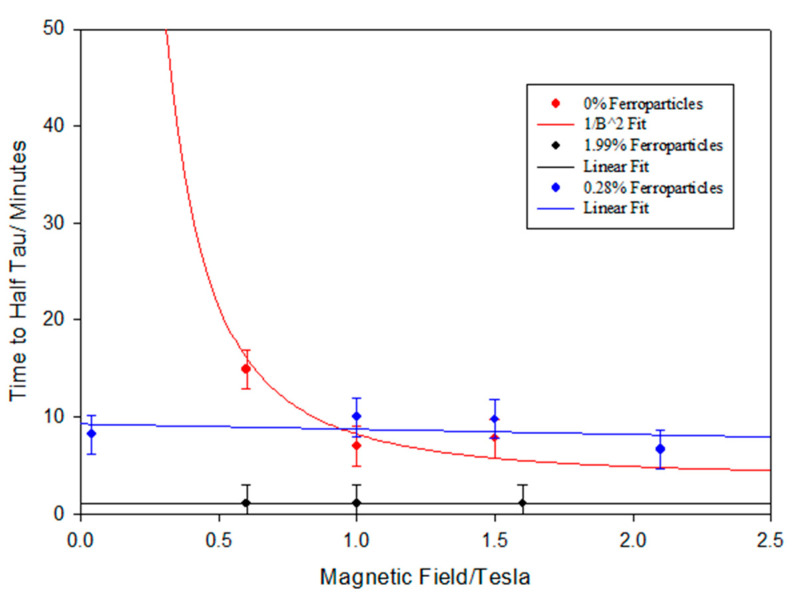
A plot of the alignment time τ for the formation of a monodomain prepared from the polymer CBZ6 alone and from the polymer plus 0.28% or 1.99%, as shown in the index insert, held at 110 °C.

**Figure 18 materials-17-05273-f018:**
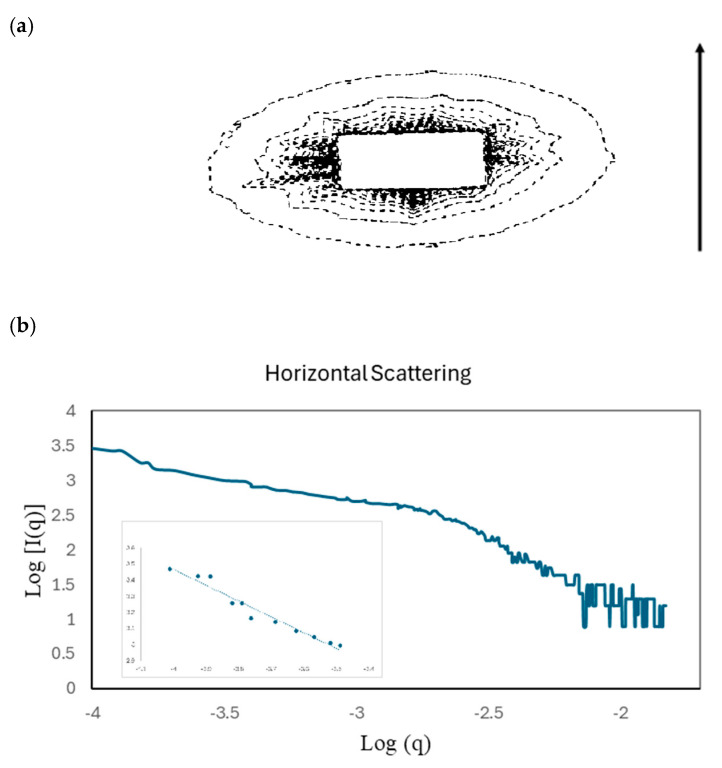
(**a**) A contour plot of the 2D small-angle X-ray scattering pattern of the monodomain containing CBZ6 plus 0.28% ferroparticles. The arrow indicates the direction of the applied magnetic field. (**b**) Log–log plot of a horizontal section of the 2D small-angle X-ray pattern for the same monodomain (CBZ6 plus 0.28% ferroparticles); the inset shows the slope at a low q, which is calculated as being −0.98.

**Figure 19 materials-17-05273-f019:**
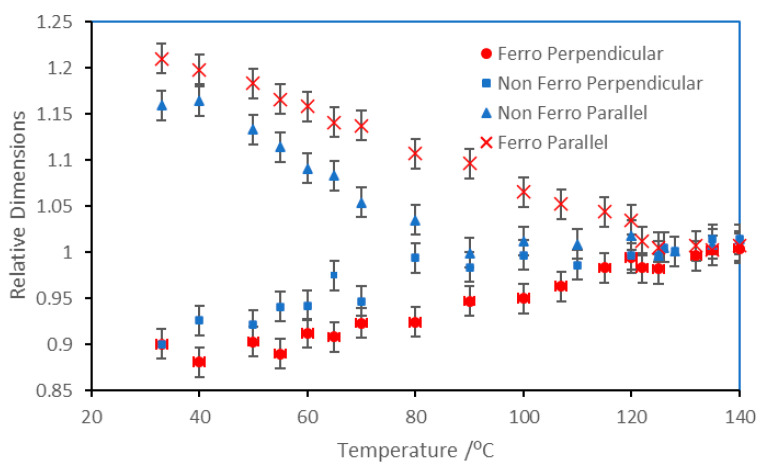
Thermally induced shape variations of monodomain liquid crystal elastomers produced from CBZ6 with and without ferro nanoparticles.

**Figure 20 materials-17-05273-f020:**
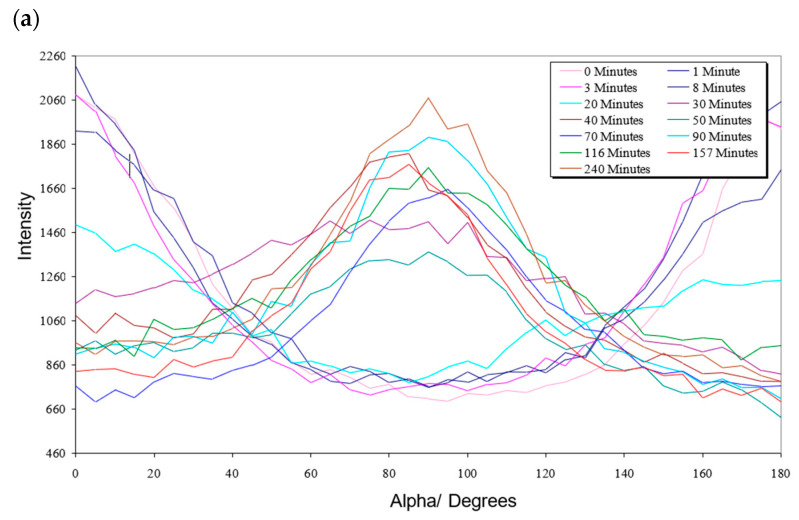
(**a**) Azimuthal scan of non-ferronematic polymer reorganization at 117 °C and 2.1 T; (**b**) azimuthal scan of 0.28% ferronematic polymer.

**Figure 21 materials-17-05273-f021:**
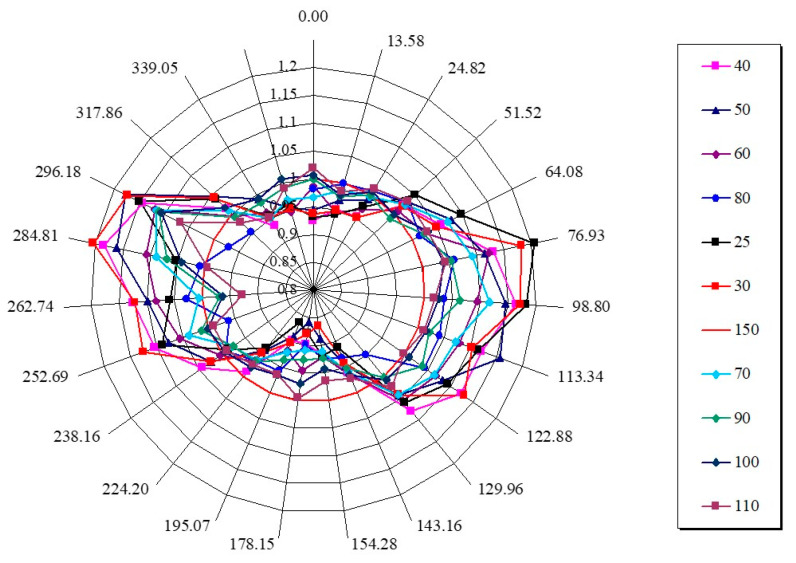
A polar plot of the sample dimensions of a monodomain prepared from the copolymer SIP 136 using a magnetic field of 2.1 T and a temperature of 110 °C.

**Table 1 materials-17-05273-t001:** Characteristics of copolymers used in this work.

Code	% I	% II	M_n_ (Daltons)	M_w_ (Daltons)	T_NI_ (°C)	T_g_ (°C)
CBZ6	94	6	2.825 × 10^4^	2.4 × 10^6^	129	33
SIP 127	94	6	1.32 × 10^4^	2.76 × 10^4^	135	33
SIP 136	94	6	1.95 × 10^4^	3.1 × 10^4^	125	33

## Data Availability

Data are available on request to the corresponding author.

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
