# Peer review of "Enhancing the Properties of Liquid Crystal Polymers and Elastomers with Nano Magnetic Particles"

_materials, 2024, doi:10.3390/ma17215273_

Round 1

Reviewer 1 Report

Comments and Suggestions for Authors

This work is concerned with the exploring of the behavior of side-chain liquid crystal polymers mixed with ferromagnetic particles in the presence of magnetic fields. It finds that at low concentrations, ferroparticles speed up the formation of a monodomain in the polymer. The alignment of ferroparticles, which form chains under weak magnetic fields, plays a crucial role, although these chains may be unstable under certain conditions. The research suggests that volume fractions below 1% yield the best results, improving properties like shape changes with temperature.  The length of the manuscript is within a regular full-length paper. The paper is interesting and could be potentially suitable for publication in Materials, but has to be improved considerably. So I suggest a major revision of the manuscript.

Reviewer 2 Report

Comments and Suggestions for Authors

The paper is a research article  on the study of the improvement of the properties of liquid crystal polymers and elastomers when magnetic nanoparticles are added to them. The authors should consider and add to the review the work on magnetocaloric nanoparticles and polymer matrices and their use for targeted drug delivery https://www.sciencedirect.com/science/article/pii/S2666032621000065?via%3Dihub In general, the paper itself is difficult to evaluate because of the terrible quality of most of the figures (unreadable inscriptions and scales), the figures are as if scanned, cut and pasted into the article from some other work.

Author Response

\see attached file

Reviewer 3 Report

Comments and Suggestions for Authors

The manuscript presents the properties of the magnetic field orientation for polymer doped with nano magnetic particles. However, the available results do not support the conclusion, more information should be provided. 

1. The chemical structure of hydroxyethyl acrylate should be shown in the manuscript. 

2. In Page 6, the authors said that "after formation of the monodomain", how to determin the formation of a monodomain structure? there are no characteration data to support the conclusion.

3. In Page 6, "...approximately 1 mm by 1.5 mm was cut from a monodomain CBZ6 elastomer with the longest length...", does the CBZ6 elastomer represent the copolymer synthesized by aromatic ester I with hydroxyethyl acrylate?

4. For Figure 5, the authors said that "Figure 5 shows a plot of the global orientation parameter against time in a magnetic field of 2.1T." in Page 6, however, Figure 5 shows a plot of the measured light vs time in fact, please check.

5. Furthermore, detailed information revealed by Figure 5 should be explained. Is the intensity in the vertical coordinate an absolute value? I don't think it can tell us the orientation information of the polymer.

6. In Figure 8, the test temprature should be indicated.

7. In Figure 8, no obvious liquid crystal birefringence and texture were shown, the change of liquid crystal texture with temperature characterized by polarizing microscope should be shown.

8. The thermal stability of polymers is one of the important indexes to evaluate the properties of materials, so thermogravimetric analysis for the coplymer and ferroparticle doped copoymers should be provided.

9. Differential scanning calorimetry is an effective method to characterize polymer phase transition behavior, so authors should offer the data for the coplymer and ferroparticle doped copoymers. 

10. Picture 5 and Picture 10 are duplicates, Picture 6 and Picture 13 are duplicates, please check.

11. Which sample is used in the Figure 6 and 13? How about other polymers?

12. In Figure 15, what is the composition of the samples SIP198 and SIP127? coplymer of ferroparticle doped copoymer with a specific concentration?

13. The resolution of Figure 16 is to low, please replace a new one.

14. A table summarizing the properties of copolymer and ferroparticle doped copoymers with different concentrations, for example, molecular weight, glass transition, phase transition, orientation parameters, etc, should be provided.

15. There are too many spelling and grammar errors in the manuscript, a comprehensive inspection and modification shoud be conducted.

16. The title shoule be reconsidered, there is no elastomers in this manuscript. 

Comments on the Quality of English Language

There are too many spelling and grammar errors in the manuscript, a comprehensive inspection and modification shoud be conducted.

Round 2

Reviewer 1 Report

Comments and Suggestions for Authors

This paper is recommended for the publication in its current version.

Reviewer 2 Report

Comments and Suggestions for Authors

all my comments have been addressed

Reviewer 3 Report

Comments and Suggestions for Authors

It can be published after the improving English expression.

Comments on the Quality of English Language

The English expression should be enhanced further.